

# An improved framework to predict river flow time series data

Hafiza Mamona Nazir[1], Ijaz Hussain[1], Ishfaq Ahmad[2], Muhammad Faisal[3,4] and Ibrahim M. Almanjahie[2]

[1] Department of Statistics, Quaid-i-Azam University, Islamabad, Pakistan
[2] Department of Mathematics, College of Science, King Khalid University, Abha, Saudi Arabia
[3] Faculty of Health Studies, University of Bradford, Bradford, United Kingdom
[4] Bradford Institute for Health Research, Bradford Teaching Hospitals NHS Foundation Trust, University of Bradford, Bradford, United Kingdom

Corresponding author
Ijaz Hussain, ijaz@qau.edu.pk

## ABSTRACT

Due to non-stationary and noise characteristics of river flow time series data, some pre-processing methods are adopted to address the multi-scale and noise complexity. In this paper, we proposed an improved framework comprising Complete Ensemble Empirical Mode Decomposition with Adaptive Noise-Empirical Bayesian Threshold (CEEMDAN-EBT). The CEEMDAN-EBT is employed to decompose non-stationary river flow time series data into Intrinsic Mode Functions (IMFs). The derived IMFs are divided into two parts; noise-dominant IMFs and noise-free IMFs. Firstly, the noise-dominant IMFs are denoised using empirical Bayesian threshold to integrate the noises and sparsities of IMFs. Secondly, the denoised IMF's and noise free IMF's are further used as inputs in data-driven and simple stochastic models respectively to predict the river flow time series data. Finally, the predicted IMF's are aggregated to get the final prediction. The proposed framework is illustrated by using four rivers of the Indus Basin System. The prediction performance is compared with Mean Square Error, Mean Absolute Error (MAE) and Mean Absolute Percentage Error (MAPE). Our proposed method, CEEMDAN-EBT-MM, produced the smallest MAPE for all four case studies as compared with other methods. This suggests that our proposed hybrid model can be used as an efficient tool for providing the reliable prediction of non-stationary and noisy time series data to policymakers such as for planning power generation and water resource management.

## INTRODUCTION

The economic development of any country is directly linked to the proper management of their water resources operations that can minimize the effects of various natural disasters such as floods and droughts. Therefore, river flow time series prediction is an imperative task, which plays a significant role for effective and appropriate water resource planning and management, early flood warning, irrigation, and hydropower generation

(*Yaseen et al., 2018a*; *Yaseen et al., 2018b*). Several algorithms have been used for prediction and estimation for river flow time series data (*Aichouri et al., 2015*; *Shathir & Saleh, 2016*; *AlMasudi, 2018*; *Gjika, Aurora & Arbesa, 2019*). The Box-Jenkins methodology technique (*Box & Jenkins, 1970*) is commonly used in the literature (*Shathir & Saleh, 2016*) because it can be used in a wide class of models i.e., Autoregressive (AR), Moving Averages (MA), Autoregressive Integrated Moving Averages Model (ARIMA), Seasonal ARIMA. But the problem with such statistical models is that they only considered the stationary and linear behavior of the process (*Di, Yang & Wang, 2014*). However, river flow time series data is non-linear, non-stationary, multi-scale and noise-corrupted (*Di, Yang & Wang, 2014*) as stochastic nature of several factors (e.g., rainfall, evaporation, and temperature). This complex non-stationary, multi-scale and noise-corrupted characteristics make the prediction a challenging task. During the past decades, this issue has been addressed by developing some data-driven models i.e., Artificial Neural Network (ANN), model tree, Support Vector Machine (SVM), Adaptive Inference-Based Neural Network (AIBNN), Extreme Learning Machine (ELM) (*Ali et al., 2018*). *Yaseen et al. (2018a)* and *Yaseen et al. (2018b)* demonstrated the viability of ELM and enhanced version of ELM method to forecast river flow data as compared to other statistical models. However, the data-driven models ignore the time-varying and noise characteristics of river flow processes which deprives the researcher to efficiently predict such data. To address the drawback of such data-driven models, several hybrid models are introduced to extract the time-varying information and reduce noises which ultimately increase the prediction accuracy of data-driven models (*Toth, Brath & Montanari, 2000*; *Di, Yang & Wang, 2014*; *Su et al., 2016*; *Kang et al., 2017*; *Hadi & Tombul, 2018*; *Yaseen et al., 2018a*; *Yaseen et al., 2018b*). The hybrid model uses data pre-processing methods such as Singular Spectrum Analysis (SSA), Wavelet Analysis (WA), Empirical Mode Decomposition (EMD) and Empirical-EMD (EEMD) with data-driven models also called intelligence models. An advantage of such data decomposition methods is that they used not only for decomposing the data into time-frequency components but also used to separate noises from data. *Wu & Chau (2011)* coupled data pre-processing techniques i.e., SSA with ANN to accurately model the rainfall and river flow data. *Azadeh et al. (2011)* demonstrated the ability to use the data pre-processing method to enhance the precision of data-driven models. They used various data processing techniques and reported that the processed non-linear data is efficiently forecasted with simple statistical and intelligent models. *Gjika, Aurora & Arbesa (2019)* found that proper consideration of the volatility nature of non-linear data through data pre-processing methods could improve the prediction quality. *Santos et al. (2019)* proposed a model by coupling WA and ANN techniques. They used WA to transform the daily flow time series data to enhance the precision of the ANN model. The data pre-processing methods used to decompose the non-stationary and non-linear data into physical modes of time-frequency components (*Han & Liu, 2009*; *Azadeh et al., 2011*; *Wang et al., 2015*; *Peng et al., 2017*). The derived time-frequency components, also called multi-scale components (*Nazir et al., 2019*), are predicted through a mixture of data-driven models accurately (*Azadeh et al., 2011*). Further, the time-frequency components, are filtered out by using appropriate thresholds. The reason for using a filter is to preserve significant features of original time

series data while removing noises or sparsity from multi-scale components. A growing body of research on denoising found that the extracted high and low multi-scale components in different fields can be found by using linear and non-linear thresholds. Moreover, many other methods of thresholds including the Stein Unbiased Risk Estimator (SURE) (*Candes, Sing-Long & Trzasko, 2012*; *Hansen, 2017*), the fixed and soft threshold (*Di, Yang & Wang, 2014*; *Nazir et al., 2019*), and minimax algorithms (*Hansen, 2017*) are also used to remove noises and to preserve important information from complex data. *Peng et al. (2017)* developed a novel hybrid model by employing an empirical wavelet transform estimator to remove the redundant noises from river flow data. Further, the denoised data are predicted through Particular Swarm Optimization based-ANN model (PSO-ANN). They demonstrated the efficiency of their proposed model over simple PSO-ANN model without denoising. *Di, Yang & Wang (2014)* proposed a hybrid model by considering two data pre-processing methods i.e., EMD and WA with a soft and hard threshold to find the denoised time-varying information to decrease the complexity of hydrological series. *Holzfuss & Kadtke (1993)* suggested that the noise reduction methods coupled with the radial basis functions may enhance the quality of traditional statistical and data-driven models.

However, WA-, EMD-, and EEMD-based noise removal techniques have their own drawbacks in extracting the optimal multi-scale components. WA-based hard and soft threshold first comprised of the choice of mother wavelet, which is subjectively selected among many wavelet basis functions. The subjective selection of mother wavelet may also cause errors which decrease the performance of hard and SURE thresholds. Moreover, the EMD is a purely data-driven technique, used for pre-processing data, which is effected through its own mathematical property of mode mixing problem resulting in spurious time-frequency information. To overcome the drawback of EMD, the EEMD is introduced which is an improved version of EMD to solve the mode mixing problem. The EEMD added Gaussian white noise successively to solve noise-assisted (*Hadi & Tombul, 2018*; *Yaseen et al., 2018a*; *Yaseen et al., 2018b*). Although an improved EEMD has proved useful for denoising hydrological time series data (*Jiao, Guo & Ding, 2016*), it also has some drawbacks that may deprive the researchers in extracting accurate multi-scale components i.e., Intrinsic Mode Function (IMF) by simple averaging them. However, to cope with the simple averaging problem of EEMD, Complete Ensemble Empirical Mode Decomposition with Adaptive Noise (CEEMDAN) have been proposed by *Torres et al. (2011)*. The application of CEEMDAN is successfully applied to derive the time-scale components of hydrological time series data (*Antico, Schlotthauer & Torres, 2014*). *Johnstone & Silverman (2005)* have considered an empirical Bayesian approach to threshold the multi-scale components derived from WA. They argued that the empirical Bayesian method efficiently models the sparsity and noises of complex data by considering multiple priors for each level. One would hope such methods which possibly estimate thresholds that stably reflect the noises from sparse multi-scale to enhance the accuracy and reliability of prediction performance of complex river flow data.

In this study, we aimed to improve denoising stage of the hybrid model by the novel framework i.e., CEEMDAN-Empirical Bayesian Threshold estimator (EBT) estimator

to optimally reduce noises from IMFs which are further used as inputs to get a precise prediction of river flow data which plays a decisive role in the accurate prediction. The principal motivation of choosing EBT is that it is purely a data-based method which deals different level of noises efficiently because of some high multi-scale components derived from CEEMDAN relatively sparse than the lower time-scale components.

## PROPOSED METHOD

The proposed method is comprised of four stages such as decomposed, denoised using novel threshold, prediction, and aggregation. The CEEMDAN method is used as a decomposition tool to handle the non-linear and non-stationary data by extracting IMFs. The extracted IMFs are further divided into two parts; one part is comprised of noisiest IMFs which contains errors and sparsity and the second part is noise free IMFs. The noisiest IMF's coupled with novel Empirical Bayesian Threshold (EBT) estimator to get a ride from noises and sparsity to denoise IMF's. Then the denoised IMF's are predicted through the complex data-driven model and noise free IMF's are predicted through a simple stochastic model. Finally, all the predicted IMF's are aggregated to get the final prediction. The proposed framework of deriving multi-scale IMF's through CEEMDAN coupled with optimal denoising method i.e., EBT plays a vital role in the accurate prediction of river flow data. To our convenience, the proposed strategy is labialized as; CEEMDAN (decomposed), EBT (denoised) and Multi Models MM (data-driven and stochastic model) i.e., CEEMDAN-EBT-MM and the proposed scheme is illustrated in Fig. 1.

### Decomposition stage

*EMD:* To decompose non-linear and non-stationary data, the EMD method introduced by *Huang et al. (1998)* which decomposed data into IMF's by satisfying two conditions as follows: **(a)** The number of zero crossings and extreme, from complete data, must be equal or differ to one at most; **(b)** At all levels, mean value of envelope must be zero.

The complete EMD procedure is defined as:

1. Find all local maxima and minima from data $y(t)$, ( $t = 1, 2, .., N$). Use cubic splines interpolation to find an upper envelope from maxima $e_{\max(t)}$ and lower envelope from minima $e_{\min(t)}$.

2. Calculate the average of upper and lower envelope $m(t) = (e_{\max(t)} + e_{\min(t)})/2$. Take the deviations between original time series data and calculated envelope mean as:

$$g(t) = y(t) - m(t) \tag{1}$$

3. Match the requirements of $g(t)$ which is defined in **(a)** and **(b)** as an IMF, if the conditions are satisfied then mark this $g(t)$ as $i$th IMF.

4. In the next step replace original time series $y(t)$ by $r(t) = y(t) - g(t)$, if $g(t)$ does not meet the **(a)** and **(b)** then just replace $y(t)$ with $g(t)$.

5. The process of (1–4) is repeated until no IMF is being extracted from the residue $r(t)$.
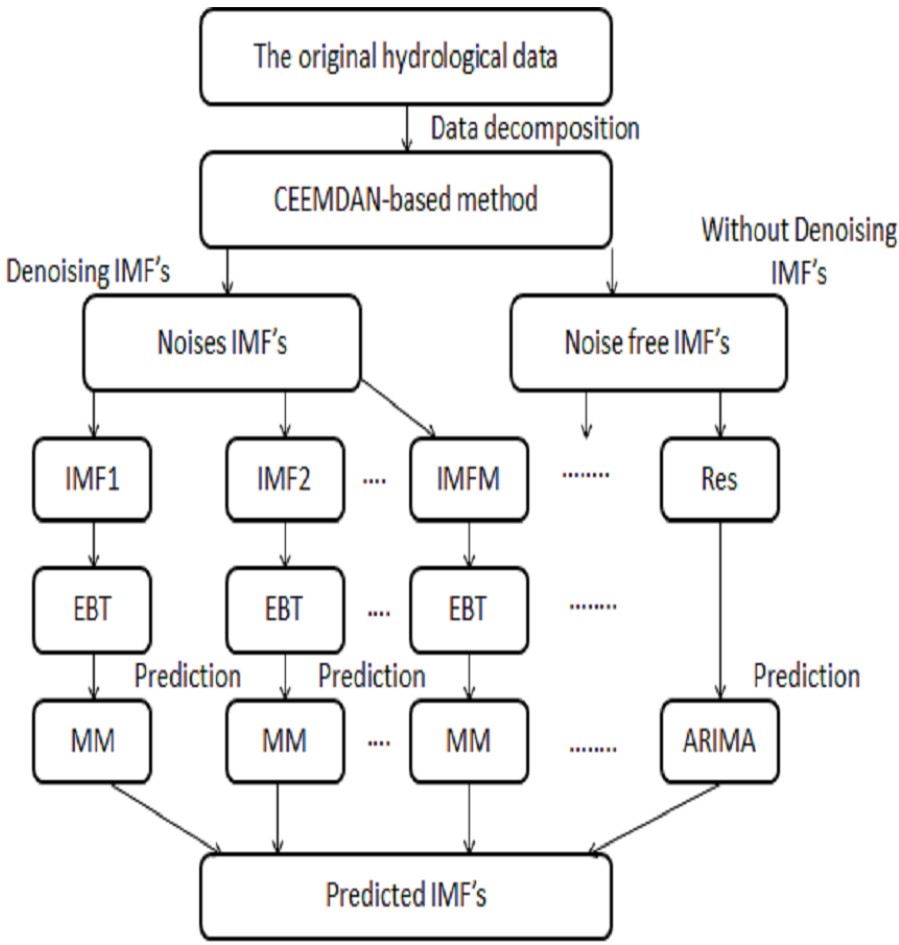

**Figure 1** **The proposed CEEMDAN-EBT-MM structure to predict non-linear river flow data.**

In the end, original time series data will be written as the summation of all extracted IMF's and residue as:

$$y(t) = \sum_i^l g_i(t) + r(t) \qquad (2)$$

where $g_i(t)$ is the $i$th IMF and $r(t)$ is the trend of the signal. However, EMD techniques have two drawbacks. The one is the endpoint problem that is the extreme values of two ends of the series can't be determined properly which distort the IMF's and the other is mode mixing aliasing in which same IMF contains even more than two frequencies.

*EEMD:* To resolve the mode mixing problem of EMD, (*Wu & Huang, 2004*) proposed EEMD. The EEMD decompose the non-linear signals into IMF's as follows:
a)  Add a white Gaussian noise series to the original data set as follows:

$$y_2(t) = y(t) + n(t) \qquad (3)$$
b)  Decompose the new $y_2(t)$ with EMD and obtains the IMFs.

c) Repeat step (a) and (b) $m$th time i.e., $(j = 1,2,...,m)$ with different white noises to get IMF's from new series.

d) Find the ensemble means of all IMFs obtained as $m$thensemble time as follows: where $k = 1,2,..,K$ is $k$th IMF.

$$\overline{imf}_k = \sum_{j=1}^{m} IMF_{jk}^{m}/m \tag{4}$$

where $k = 1,2,..,K$ is $k$th IMF.

*CEEMDAN*: Although the mode mixing problem is alleviated with EEMD, taking the simple averages of IMF's without considering the independent addition of white noises could not completely remove noises from IMF's. To tackle the simple averaging problem of EEMD, the CEEMDAN function is introduced by *Torres et al. (2011)*. Here in our study, CEEMDAN is used to decompose the river flow data which is briefly described as follows:

1) The CEEMDAN add Gaussian noises like EEMD in signals as follows:

$$y_3(t) = y(t) + w_0 n^j(t) \tag{5}$$

where $w_0$ is the amplitude of the added white noises and $(j = 1,2,...,m)$. Find the first IMF using simple EEMD defined as:

$$\widetilde{IMF}_1 = \sum_{j=1}^{m} IMF_{j1}^{m}/m \tag{6}$$

2) Compute the deviation of original signals from the first IMF as:

$$r_1(t) = y(t) - \overline{imf}_1 \tag{7}$$

3) Decompose $r_1(t) + w_0 n^j(t)$ to get first IMF and find the second IMF as:

$$\widetilde{IMF}_2 = \sum_{j=1}^{m} IMF_{j1}^{m}/m \tag{8}$$

4) Repeat the (2–3) until stoppage criteria are met and the residual contains not more than two extremes. Finally, the residual is defined as:

$$R(t) = y(t) - \sum_{k=1}^{K} \widetilde{IMF}_k \tag{9}$$

However, the selection of a number of ensemble and amplitude of white noise is still an open challenge but here in our study we used a number of ensemble members as 100 and standard deviation of white noise is settled as 0.2 according to *Di, Yang & Wang (2014)*.

*Identification of noisiest IMFs:* After deriving the IMF's, next step is to screen out the noise only IMF's and noise free IMFs (*Wei et al., 2016*). Two types of IMF's are derived from CEEMDAN/EEMD; the IMF's contains high frequency which is corrupted with sparsity and noises (*Wei et al., 2016*) and the second part of IMF's comprised on low frequencies which are free from noises (*Wei et al., 2016*). To get numerical validation, cross-correlation is calculated between all extracted IMF's and original river flow data. The low cross-correlation implies that the high-frequency IMF's are overwhelmed with noises.

Then, the noise only IMF's are further denoised through the appropriate threshold to get the important features from them and to get rid of noises.

## Denoising stage

After decomposing the non-linear and non-stationary data into IMF components, an appropriate estimator is chosen to remove noises and sparsities from extracted IMFs. The reason for selecting an appropriate estimator is to find an optimal value of threshold as the highest threshold value would lead to biases, whilst the lowest threshold value would increase the noise variance. The IMFs which are extracted are mostly empirical Bayesian threshold estimator is adapted to denoised the noisiest IMFs. Later, the existing soft and hard threshold and improved threshold functions are used for comparison purposes. Details of all estimators are given below:

*CEEMDAN/EEMD-based Empirical Bayesian Threshold:* To estimate the sparseness and noises from decomposed IMF's (*Wei et al., 2016*), an Empirical Bayes Threshold (EBT) inspired from wavelet denoising (*Johnstone & Silverman, 2005*; *Jansen & Bultheel, 2014*) is used. For the successful application of EBT, first, all the data is scaled transformed to efficiently select the prior distribution for noises and sparsities. After the scaled transformation data follows $N(\theta_i, 1)$. Then a mixture of priors for $\theta_i$ are considered as follows:

$$f_{prior}(\theta) = (1-w)\delta_0(\theta) + w\gamma(\theta) \tag{10}$$

where $\delta_0(\theta)$ is a zero part of scaled data and $\gamma(\theta)$ is a density of non-zero part. The density of prior should be chosen in such a way that it must belong to a family of distributions whose tails decays at polynomial rates. The parameters and weights of a mixture of prior distributions are estimated through maximum likelihood approach. The reason for using this method to estimate unknowns is that it estimates weights and parameters to be proportional to the likelihood function evaluated at the estimators based on data (*Hossain, Kozubowski & Podgórski, 2018*).

Finally, the posterior median $\tilde{\theta}_i(imf, w)$ is calculated from a mixture of prior distribution is given as follows:

$$\tilde{F}_1(\mu|imf) = \int_{\mu}^{\infty} f_1(\mu|imf) d\mu \tag{11}$$

which is used as a threshold rule for $\bar{\mu}$ given data (*Johnstone & Silverman, 2005*). In general, an estimation rule comprised on $\eta(imf, t)$ defined for all $t > 0$, is a thresholding rule if and only if for all $t > 0$, $\eta(imf, t)$ is an antisymmetric and increasing function of data and $\eta(imf, t) = 0$ if and only if $|imf| \leq t$ where $t$ is defined as a median value which is calculated through the Eq. (11).

*Other traditional threshold estimators:* To compare the EBT estimator with other non-linear estimators to suppress the noises and sparsities from noisiest IMFs, Soft Threshold (ST), Hard Threshold (HT) and Improved Threshold Function (ITF) are used which are most widely used in literature (*Jeng et al., 2007*; *Candes, Sing-Long & Trzasko, 2012*; *Jansen & Bultheel, 2014*) listed as follows, respectively;

$$IMF'_{t,k} = \begin{cases} IMF_{t,k} & |IMF_{t,k}| \geq T_k \\ 0 & |IMF_{t,k}| < T_k \end{cases} \tag{12}$$

$$IMF'_{t,k} = \begin{cases} sgn(IMF_{t,k})\left(|IMF_{t,k}| - T_k\right) & |IMF_{t,k}| \geq T_k \\ 0 & |IMF_{t,k}| < T_k \end{cases} \tag{13}$$

and

$$IMF'_{t,k} = \begin{cases} sgn(IMF_{t,k})\left(\dfrac{|IMF_{t,k}| - T_k}{\exp\{3\alpha\left(\frac{IMF_{t,k} - T_k}{T_k}\right)\}}\right) & |IMF_{t,k}| \geq T_k \\ 0 & |IMF_{t,k}| < T_k \end{cases} \tag{14}$$

where, $T_k$ is the threshold value calculated as $T_k = a\sqrt{2E_k \ln(N)}$, where $k = 1,2,\ldots.,K$ and $a$ is constant takes the values between 0.4 to 1.4 with a step of 0.1 and $E_k = median(|IMF_{t,k}|, t = 1,2,..,N)/0.6745$ is median deviation of $k$th IMF.

## Prediction and aggregation

In the prediction stage, the decomposed-denoised IMFs of river flow data are predicted through some data-driven and statistical models. Specifically, the denoised IMFs are predicted through a data-driven model whenever noise-free IMFs and the residual are predicted through a simple stochastic model. To train the model, 70%, 80%, and 90% data are used, and the remaining 30%, 20% and 10% data are used to test the accuracy of the models. The selected models are briefly described as follows:

*The denoised-IMF prediction with the neural network:* A data-driven technique has been proved a powerful tool to model complex non-linear data (*Campisi-Pinto, Adamowski & Oron, 2012*). The Multi-Layer Perceptron (MLP) which is the most popular sub-model of NN (*Talaee, 2014*; *Ali et al., 2017*) consists of three layers of nodes used here for the prediction of denoised IMFs of river flow data. The complete layout of MLP is given in *Talaee (2014)*. The structure of MLP comprised of NN structure. Three nodes are included in MLP as an input layer, hidden layer, and an output layer. First, the single output is calculated by using linear combinations of inputs which are further transferred to some non-linear activation functions mathematically defined as by

$$y = \varphi\left(\sum_{i}^{n} w_i x_i + b\right) \tag{15}$$

where $w_i$ are the weights of inputs, $x_i$ are the inputs, $b$ is the biased value for each layer and $\varphi$ is a non-linear activation function which supplies output to the next layer. The mostly used activation function i.e., logistic function is used as activation function defined as follows:

$$\frac{1}{(1 + e^{-\sum_{i}^{n} w_i x_i + b})} \tag{16}$$

For the optimization of neurons, the supervised learning algorithms called back-propagation and forward-propagation can be used.

*The noise-free IMF prediction with ARIMA model:* To predict the noise-free IMF's and the residual, an autoregressive moving average model (ARMA) is selected which is described as follows:

$$IMF_t^k = \alpha_1 IMF_{t-1}^k + \ldots + \alpha_p IMF_{t-p}^k + \varepsilon_t^k + \beta_1 \varepsilon_{t-1}^k + \ldots + \beta_q \varepsilon_{t-q}^k \qquad (17)$$

where, $IMF_t^k$ is the $k$th IMF, $IMF_{t-p}^k$ is $p$th lag value of $k$th IMF $\varepsilon_t^k$ is the residual of $k$th IMF, $p$ and $q$ are autoregressive and moving average lags. Moreover, in some cases time series data is not stationary. To make such stationary, differences at an appropriate degree are used (*Box & Jenkins, 1970*). If such a situation occurs, then the model is known as ARIMA $(p, d, q)$ where $d$ is the differenced value used to make the non-stationary data stationary.

# CASE STUDY AND EXPERIMENTAL DESIGN

## Selection of study area

In this research, the largest water system of Pakistan is considered for the application of the proposed strategy. The Indus Basin System (IBS) is the largest river of Pakistan and it plays an important role specifically in power generation and irrigation department. The major tributaries of IBS i.e., River Jhelum, River Chenab, and River Kabul are selected for the present study. Pakistan is facing a large amount of frequent river flooding each year due to monsoon rain and melting snow or glaciers. As in Pakistan, glacier-covered 13,680 km$^2$ area which is estimated 13% of the mountainous areas of Upper Indus Basin (UIB). Melted water from these 13% areas adds the significant contribution of water in these rivers. which leads to complex characteristics in river flow data. Therefore, for sustainable economic development and efficient water resources planning, there is a need for analyzing such complex characteristics and predict the behaviors of river flow data at IBS and its tributaries.

## Data

To investigate the improved framework, four daily river flow data comprised on (1st-January to 19th-June) for the period of 2015–2018 is used in this study. The main river flow of Indus at Tarbela is considered with its three principal tributaries: Jhelum at Mangla, Chenab at Marala and Kabul at Nowshera. Data is measured in 1,000 ft/s. The described river flow data was obtained from the website of Pakistan Water and Power Development Authority (WAPDA).

## Evaluation criteria

*Evaluation of noise reduction methods:* The performances of denoised series needs comprehensive evaluation after using appropriate noise reduction threshold methods. In this research, to check the performances of CEEMDAN/EEMD-ST CEEMDAN/EEMD-HT, CEEMDAN/EEMD-ITF, and proposed framework i.e., CEEMDAN/EEMD-EBT, Signal-to-Noise Ratio (SNR), Mean Square Error (MSE) and Mean Absolute Error (MAE)

(*Nazir et al., 2019*) are employed which are given as follows respectively;

$$SNR = 10\log 10\left(\frac{\sum_{t=1}^{N}(y_{ot})^2}{\sum_{t=1}^{N}(y_{pt}-y_{ot})^2}\right) \tag{18}$$

$$MSE = \sqrt{\frac{\sum_{t=1}^{N}(y_{ot}-y_{pt})^2}{N}} \tag{19}$$

and

$$MAE = \frac{\sum_{t=1}^{N}|y_{ot}-y_{pt}|}{N} \tag{20}$$

where $y_{ot}$ is the $t$th observed value and $y_{pt}$ is the $t$th predicted value. Moreover, the performance of proposed model (i.e., CEEMDAN-EBT-MM) and all other models including CEEMDAN/EEMD-ST-MM, CEEMDAN/EEMD-HT-MM, CEEMDAN/EEM-ITF-MM and EEMD-ITF-MM are compared using three popular statistical measures: MSE, MAE defined in Eqs. (19)–(20) and Mean Absolute Percentage Error (MAPE) given as follows;

$$MAPE = \frac{|y_{mo}-y_{mp}|}{|y_{mo}|} * 100 \tag{21}$$

where, $y_{ot}$ and $y_{pt}$ is defined above, $y_{mo}$ is the mean value of observed values, $y_{mp}$ is the mean value of predicted values and $y_{sp}$ is the standard deviation of predicted values.

## RESULTS

This section provides results of proposed CEEMDAN-EBT-MM model and benchmark models i.e., CEEMDAN/EEMD-ST-MM, CEEMDAN/EEMD-HT-MM, CEEMDAN/EEM-ITF-MM, and EEMD-ITF-MM in steps as follows:

*Results of decomposition stage:* First, to confirm the non-stationarity of river flow data, Augmented Dickey Fuller (ADF) unit root test (*Khalili et al., 2013*) is used for all selected case studies. The test is applied to data by taking the log in order to confirm non-stationarity. Results of ADF showed that all selected river flow data i.e., Indus river flow, Jhelum river flow, Chenab river flow, and Kabul river flow is significantly non-stationary with *p*-value as 0.3353, 0.4135, 0.333 and 0.414 respectively. Then, the non-linear and non-stationary data decomposed into different time scale oscillation called IMFs to reduce the non-stationarity by extracting the time-varying characteristics of daily river flow data from all selected four stations. The CEEMDAN decomposition technique is used to extract IMF's of river flow data. All selected river flow data is decomposed into thirteen IMFs and one residual. The starting IMF's represents the highest frequencies whereas the last half IMF's showed the low frequencies and the residual represent the overall trend. The decomposed results of Indus River system is depicted in Fig. 2. The amplitude of white noise is set 0.2 as in *Di, Yang & Wang (2014)* and numbers of ensemble members are settled as 1,000. By inspecting the IMFs, it is noticed that each IMF component represents the oscillation characteristic

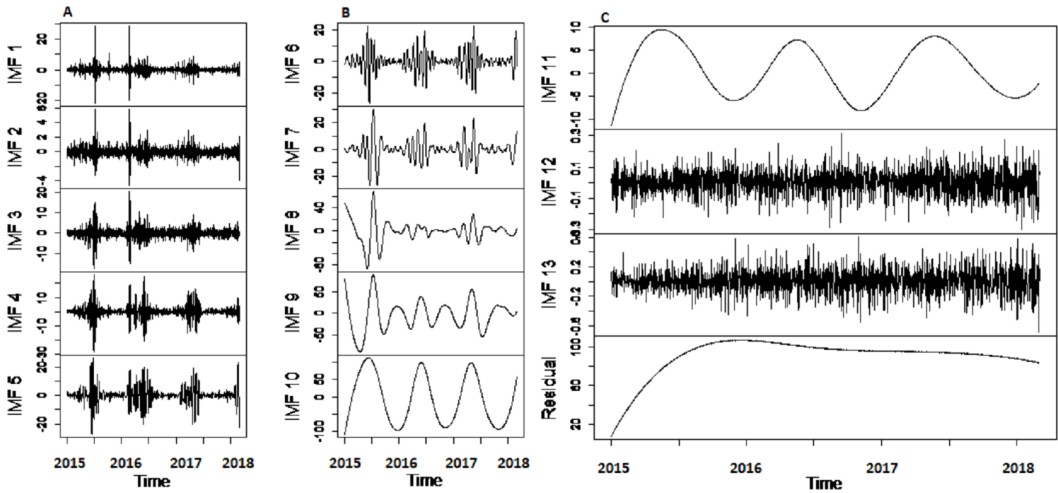

**Figure 2** **The CEEMDAN based decomposition of river flow data.** The graph (A) shows the highest frequencies and the graph (B) shows the lower frequencies and graph (C) shows irregular variations with the trend.

Further, the cross-correlation method is employed to find the noisiest IMFs from all thirteen IMFs. To do this, first the decomposed IMFs are further divided into two parts to find the noisiest IMFs by using the cross-correlation between IMFs and original river flow data. The low correlation implies the more uncertainty present in IMFs. The first ten IMFs showed the least correlation with original river flow data indicated that these IMFs are overwhelmed with noises. The graph of cross-correlation between Indus river flow data and first IMF and between Indus river flow and eleventh IMF is depicted in left and right corner of Fig. 3, which shows that the first IMF is filled with noises with low correlation at all lags and eleventh IMF is free from noises as it showed 0.75 correlation not only at lag zero but also at other lag values. For all four rivers, first tenth IMFs are characterized as noisiest IMFs and last three IMFs are labeled as noise free IMFs.

*Results of denoising stage:* the next step is to denoise the noisiest and sparse IMFs.

To eliminate noises from IMFs, EBT estimator is used as a filter which assumes a mixture of prior as defined in Eq. (10), for each IMF separately by considering the nature of IMFs. First, the scaled transformation is applied to get normal distribution so that each IMF follows $N(\theta_i, 1)$. According to the nature of IMFs as depicted in Fig. 4, it is known that most of the coefficients in all IMFs are zero and some are non-zero in which fewer coefficients are either very low or high. By looking (Fig. 4) both zero and non-zero part of IMFs, a mixture of an atom of probability at zero and multiple distributions are considered for non-zero part (*Johnstone & Silverman, 2005*). Among all of them, Laplace distribution is configured out as prior distribution of $\theta_i$ with maximum SNR. Finally, the important coefficients of IMFs are preserved with posterior median threshold estimator described in Eq. (11) by attaining highest SNR value with minimum MSE and MAE given in Table 1 for all selected case studies. For comparison purpose, the conventional denoising methods as ST, HT and ITF are implemented on all river flow data. We observed that SNR of CEEMDAN-EBT

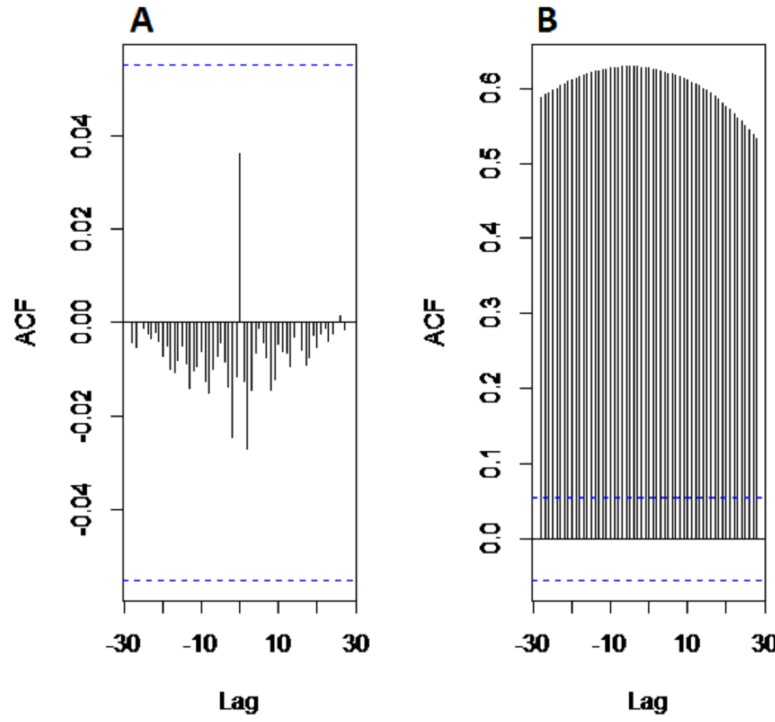

**Figure 3** **The cross-correlational graph.** The plot (A) shows the cross-correlation between first IMF and original Indus river flow data and plot (B) shows the cross-correlation between eleventh IMF and original Indus river flow.

based decomposed and denoised method is larger than all other CEEMDAN/EEMD-ST, CEEMDAN/EEMD-HT and CEEMDAN/EEMD-ITF based methods as they don't consider the sparsity and magnitude of noises separately to remove noises from data. The reconstructed denoised graph of proposed CEEMDAN/EEMD-EBT and bench mark models CEEMDAN/EEMD-HT,CEEMDAN/EEMD-ST and CEEMDAN/EEMD-ITF for Indus and Jhelum are shown in Figs. 5/6 respectively. From Figs. 5 and 6, it is shown that CEEMDAN/EEMD-ST over estimated noises for Jhelum, Chenab and Kabul river flow and, the performance of CEEMDAN/EEMD ITF is worst for Indus river flow however, the proposed CEEMDAN-EBT based model shown optimal performance for all case studies.

*Results of prediction stage:* The decomposed and denoised IMFs of all selected case studies are further predicted through data-driven and statistical model. The denoised IMFs are predicted through MLP-neural network model. Training is performed by using forward and back-propagation by setting the learning rate parameter between 0.1 to 1. The back-propagation method with the optimal learning rate is selected to test the model. The second part of the decomposed IMFs comprised of noise-free IMFs and the residual, which are predicted through simple traditional statistical model (i.e., ARIMA (p, d, q)) for all case studies. The river flow data of all four rivers are splitted, 70%, 80% and 90% for the training set and 30%, 20% and 10% for testing set. The results achieved by such splitting criteria are not much deviated from each other so the only values of 80% training
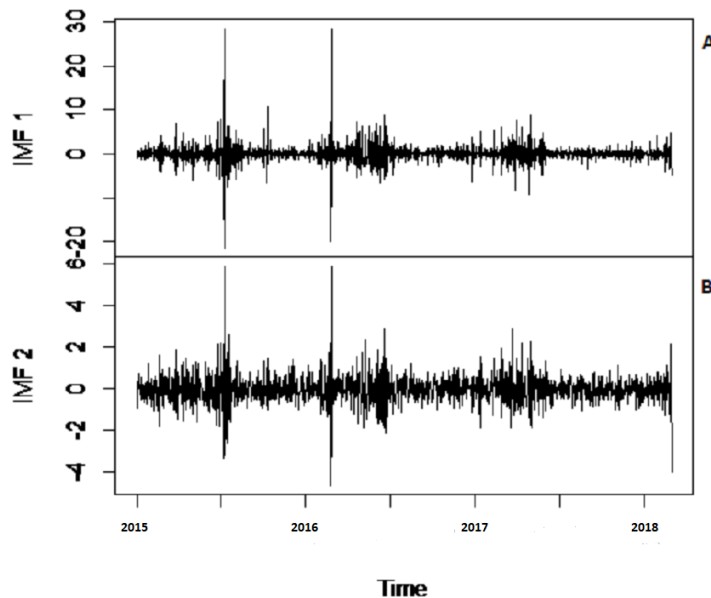

**Figure 4 The CEEMDAN based first two IMFs for Indus river inflow.** The CEEMDAN based (A) first and (B) second IMF for Indus river flow.

errors are given in Table 2. After a successful estimation of each IMF and residual, the accuracy is measured with MAE, MAPE, and MSE. The training results of proposed models with a comparison to all other models for all four river flow i.e., Indus flow, Jhelum flow, Chenab flow, and Kabul flow are presented in Table 2. The results of the proposed model i.e., CEEMDAN-EBT-MM, demonstrate its effectiveness by showing minimum MAD, MAPE and RMSE values for Indus river flow. For Jhelum and Chenab, the proposed CEEMDAN-EBT-MM model shown least MAE, and MAPE values and by attaining lower MAPE value for Kabul river flow comparative to all other methods. The predicted graph of proposed CEEMDAN-EBT-MM model and EEMD-EBT-MM with their benchmark models for Indus and Jhelum river flow are shown in Figs. 7 and 8 respectively.

## DISCUSSION

*Decomposition and denoising results:* In order to understand the applicability of our proposed CEEMDAN-EBT model, the other EEMD-based decomposition method and three different denoising methods (i.e., EEMD-HT, EEMD-ST and EEMD-ITF were employed. The evaluation of denoising of proposed CEEMDAN-EBT and benchmark models were carried out using SNR, MSE and MAE measures for all river flow data. From Table 1, it is clear that overall decomposition based on CEEMDAN performs well with improved denoising method (i.e., CEEMDAN-EBT) that efficiently eliminating noises by considering the mixture of priors for IMFs with highest SNR value and lowest MSE and MAE values; however, other existing denoising methods (i.e., CEEMDAN-HT, CEEMDAN-ST, and CEEMDAN-ITF) performed low with low SNR value and high MSE values. Moreover, comparative to CEEMDAN, the other EEMD based decomposition and

**Table 1 The statistical measures of proposed CEEMDAN-EBT and existing denoising methods for all case studies.** The results of our proposed method are indicated in bold.

| River Inflow | Method | SNR | MSE | MAE |
|---|---|---|---|---|
| Indus Inflow | CEEMDAN-HT | 15.4996 | 0.9924 | 0.7905 |
| | CEEMDAN-ST | −18.4131 | 3.1607 | 1.3692 |
| | CEEMDAN-ITF | −15.7423 | 3.1645 | 1.3713 |
| | **CEEMDAN-EBT** | **22.0440** | **1.03563** | **0.8741** |
| | EEMD-HT | −19.32131 | 49.8554 | 5.6131 |
| | EEMD-ST | −36.7393 | 36.1929 | 4.4431 |
| | EEMD-ITF | −36.4253 | 36.1766 | 4.4442 |
| | EEMD-EBT | −14.9984 | 49.31019 | 5.6353 |
| Jhelum Inflow | CEEMDAN-HT | 4.0185 | 0.6706 | 0.7905 |
| | CEEMDAN-ST | −21.0096 | 2.4233 | 1.1978 |
| | CEEMDAN-ITF | −20.4669 | 2.4234 | 1.1985 |
| | **CEEMDAN-EBT** | **20.7036** | **0.0012** | **0.0142** |
| | EEMD-HT | −12.7805 | 3.8168 | 1.5496 |
| | EEMD-ST | −28.2531 | 5.7458 | 1.7912 |
| | EEMD-ITF | −28.0296 | 5.7463 | 1.7927 |
| | EEMD-EBT | 9.7364 | 3.2357 | 1.4390 |
| Chenab Inflow | CEEMDAN-HT | −4.8656 | 0.3252 | 0.4230 |
| | CEEMDAN-ST | −17.3340 | 3.1607 | 1.3692 |
| | CEEMDAN-ITF | −16.4194 | 1.1085 | 0.7870 |
| | **CEEMDAN-EBT** | **31.6470** | **0.0004** | **0.0165** |
| | EEMD-HT | −8.1820 | 3.8706 | 1.5152 |
| | EEMD-ST | −29.2759 | 4.9703 | 1.6133 |
| | EEMD-ITF | −29.0425 | 4.9693 | 1.6148 |
| | EEMD-EBT | −12.8752 | 3.5997 | 1.5085 |
| Kabul Inflow | CEEMDAN-HT | 10.8756 | 0.4823 | 0.4986 |
| | CEEMDAN-ST | −22.289 | 2.2647 | 1.1272 |
| | CEEMDAN-ITF | −21.6878 | 2.2649 | 1.1280 |
| | **CEEMDAN-EBT** | **31.9856** | **0.0003** | **0.0144** |
| | EEMD-HT | −2.4276 | 5.4761 | 1.7961 |
| | EEMD-ST | −32.0408 | 8.5625 | 2.0934 |
| | EEMD-ITF | −31.8501 | 8.5601 | 2.0949 |
| | EEMD-EBT | −10.2153 | 5.1512 | 1.7999 |

denoising methods i.e., EEMD-EBT, EEMD-ST, EEMD-HT, and EEMD- ITF for all case studies are also performed poor as the EEMD based decomposition exhibits clear mode mixing as shown in Fig. 6, where results for Indus and Jhelum river data are plotted. The MSE and MAE values of EEMD-EBT, EEMD-ST, EEMD-HT, and EEMD- ITF methods having SNR values are very low and MSE and MAE values high due to poor decomposition and denoising methods for all four rivers data which implies that CEEMDAN has the ability to optimally extract the IMFs which are further processed with optimal denoising methods to get smooth noise free IMFs as shown in Fig. 5, where denoised results of Indus and Jhelum river flow are plotted. From the results shown in Table 1, it is concluded that our

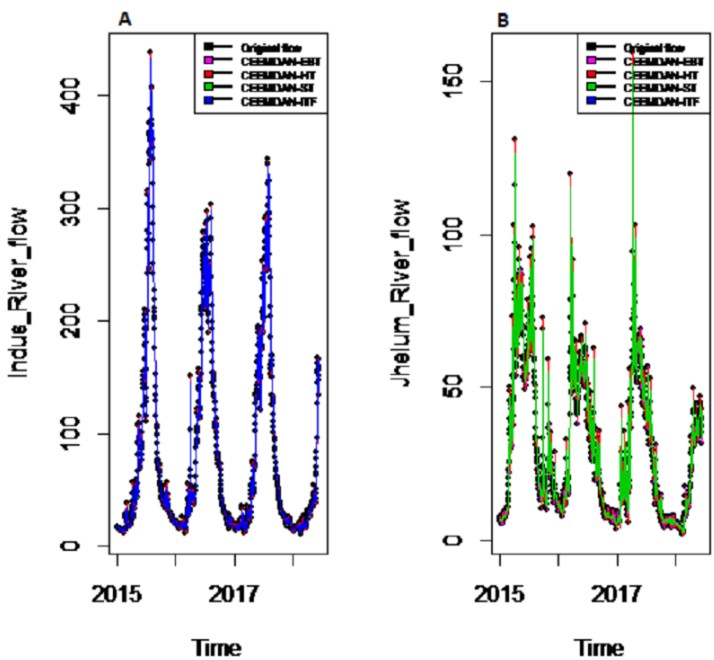

**Figure 5** The CEEMDAN based decomposed and denoised series of proposed EBT and existing ST, HT and ITF of (A) Indus and (B) Jhelum river flow.

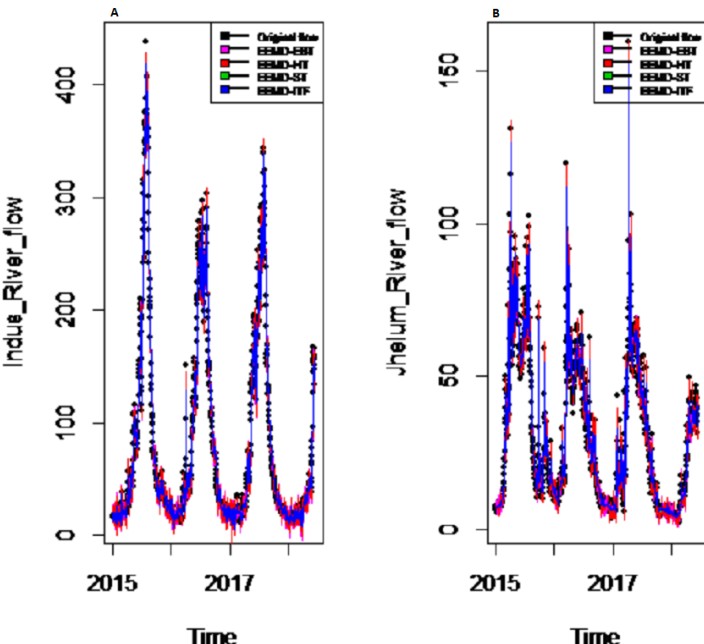

**Figure 6** The EEMD based decomposed and denoised series of (A) Indus and (B) Jhelum rivers flow.

**Table 2  Evaluation index of testing prediction error of proposed models (CEEMDAN-EBT-MM) with all selected models for all four case studies.** The results of our proposed method are indicated in bold.

| Rivers | Method | MAE | MAPE | MSE |
|---|---|---|---|---|
| Indus River Inflow | CEEMDAN-HT-MM | 9.5938 | 0.1317 | 13.7910 |
| | CEEMDAN-ST-MM | 8.7531 | 0.1413 | 17.2590 |
| | CEEMDAN-ITF-MM | 8.4614 | 0.1349 | 14.7238 |
| | **CEEMDAN-EBT-MM** | **5.6837** | **0.1071** | **9.8704** |
| | EEMD-HT-MM | 16.7567 | 5.0868 | 18.7316 |
| | EEMD-ST-MM | 12.7899 | 0.1480 | 17.2683 |
| | EEMD-ITF-MM | 12.7459 | 0.1648 | 17.7453 |
| | EEMD-EBT-MM | 17.4940 | 0.1338 | 15.4142 |
| Jhelum River Inflow | CEEMDAN-HT-MM | 6.9537 | 0.3502 | 0.0547 |
| | CEEMDAN-ST-MM | 6.0630 | 0.0577 | 0.3912 |
| | CEEMDAN-ITF-MM | 6.0558 | 0.0407 | 0.1949 |
| | **CEEMDAN-EBT-MM** | **6.0185** | **0.0400** | 0.2609 |
| | EEMD-HT-MM | 7.1364 | 0.1157 | 1.5763 |
| | EEMD-ST-MM | 8.8392 | 0.1227 | 1.7755 |
| | EEMD-ITF-MM | 6.4300 | 0.1451 | 2.4853 |
| | EEMD-EBT-MM | 7.2961 | 0.0414 | 0.2015 |
| Chenab River Inflow | CEEMDAN-HT-MM | 6.6962 | 0.0115 | 0.0140 |
| | CEEMDAN-ST-MM | 6.2191 | 0.0110 | 0.0141 |
| | CEEMDAN-ITF-MM | 6.2198 | 0.0217 | 0.0504 |
| | **CEEMDAN-EBT-MM** | **6.0027** | **0.0025** | 0.1934 |
| | EEMD-HT-MM | 6.9454 | 0.0818 | 0.7159 |
| | EEMD-ST-MM | 5.3718 | 1.0425 | 115.923 |
| | EEMD-ITF-MM | 6.5058 | 0.3943 | 16.678 |
| | EEMD-EBT-MM | 7.0002 | 0.0196 | 0.0410 |
| Kabul River Inflow | CEEMDAN-HT-MM | 8.5972 | 0.2274 | 7.0561 |
| | CEEMDAN-ST-MM | 7.8189 | 0.1676 | 3.8311 |
| | CEEMDAN-ITF-MM | 7.8192 | 0.2328 | 3.3163 |
| | **CEEMDAN-EBT-MM** | 8.5971 | **0.1660** | 3.7578 |
| | EEMD-HT-MM | 8.7354 | 0.2093 | 5.9963 |
| | EEMD-ST-MM | 6.6151 | 0.2384 | 0.2028 |
| | EEMD-ITF-MM | 6.6042 | 0.1864 | 0.2881 |
| | EEMD-EBT-MM | 9.0280 | 0.1878 | 4.8166 |

proposed strategy (i.e., CEEMDAN-EBT) performs better than that of all other methods (i.e., CEEMDAN/EEMD-ST, CEEMDAN/EEMD-HT and CEEMDAN/EEMD-ITF and EEMD-EBT based decomposition and denoising methods).

*Final prediction model:* To verify the superiority of proposed (i.e., CEEMDAN-EBT-MM) strategy to model the complex river flow data, we choose the CEEMDAN/EEMD-ST-MM, CEEMDAN/EEMD-HT, CEEMDAN/EEMD-ITF AND EEMD-EBT-MM models to analyze the prediction results of non-linear and noise-corrupted data. Our proposed prediction framework based on decomposition and novel strategy of denoising performs well as compared to all other decomposition and denoising methods. As shown in Table 2

Fast

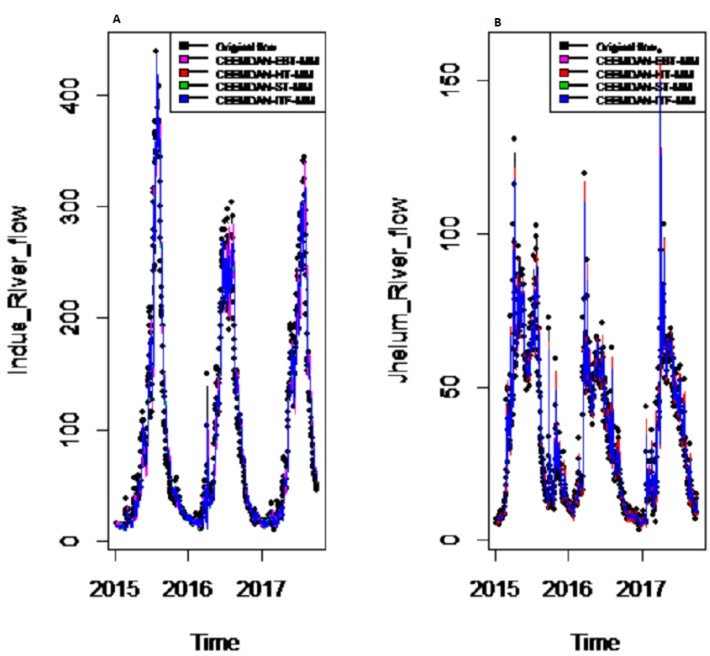

**Figure 7** Prediction results of (A) Indus and (B) Jhelum river flow using proposed CEEMDAN-EBTMM with a comparison to all benchmark models.

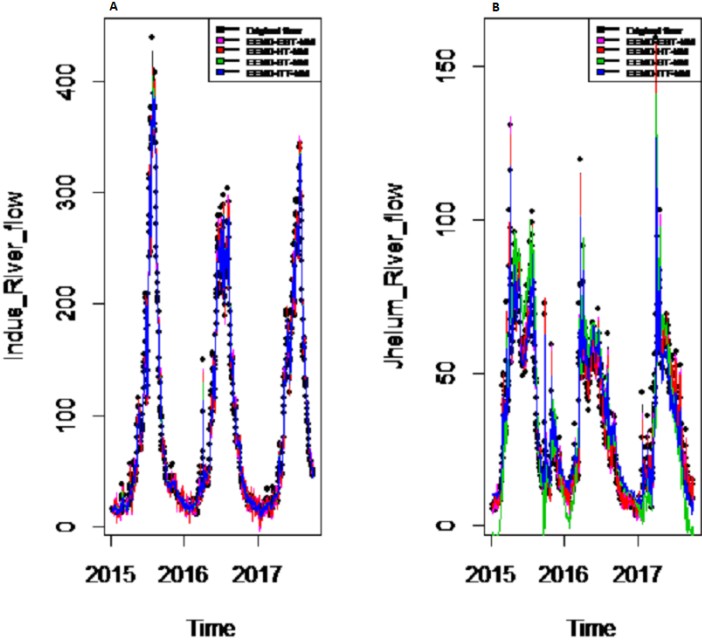

**Figure 8** Prediction results of (A) Indus and (B) Jhelum river flow using EEMD-EBT-MM with a comparison to all benchmark models.

and Fig. 7, the proposed model (i.e., CEEMDAN-EBT-MM) exhibits a good prediction performance for Indus river flow with least MAE, MSE and MAPE values. The MAE and MAPE values of Jhelum and Chenab river flow are also better than the other models. For Kabul river flow, our proposed model showed its efficiency with the least MAPE value. The other models are not consistent in terms of efficiency, their prediction behaviors for each river vary as CEEMDAN-HT-MM shows effective performance for Indus river flow but was poor in predicting Kabul river flow. However, overall the proposed CEEMDAN-EBT-MM model shows consistent and excellence prediction results which indicate that with appropriate decomposition and the novel improvement of denoising technique, one can overall improve the performance of existing data-driven models to handle the river flow data.

In this study, the river flow of the selected Indus River system exhibited seasonal persistence at several multi-scales which manifested through data decomposition method, and was used to enhance the prediction accuracy of river flow. Moreover, instead of overall season, *Kalhoro et al. (2017)* shown that there is a further sub-seasonal variation i.e., high flow, called Kharif season, and low flow, called Rabi season, present in Indus River system. The majority of Pakistan's population lives in rural areas and have the occupation of farming, which is directly influenced by the severity of both sub-seasonal variations. In the future, there is a need for establishing reliable sub-season identification and prediction methods that will be fruitful for efficient water allocation in both seasons, which can ultimately boost the economy of Pakistan.

## CONCLUSION

In this paper, due to the non-linearity and noises complexity of the river flow data, we proposed a strategy to improve the prediction of data-driven models with appropriate decomposition and a novel denoising method. Our proposed denoising method improves the performance of the CEEMDAN-based decomposition method by improving the time-scale components which enhance the prediction accuracy of data-driven models. The proposed method comprised of four steps such as decomposition, denoising and prediction, and aggregation. The performance of the proposed CEEMDAN-EBT-MM model is evaluated using four daily river flow data of IBS. The CEEMDAN-EBT-MM having the smallest MAPE values for all four case studies compared to other benchmark models. The improved results suggest that the proposed hybrid model can be used as an efficient tool for providing a reliable prediction of river flow data to policymakers for planning power generation and water resource management.

### Funding

The Deanship of Scientific Research at King Khalid University, Kingdom of Saudi Arabia funded this work through research groups program under the project number RGP-1/103/40. The funders had no role in study design, data collection and analysis, decision to publish, or preparation of the manuscript.

### Grant Disclosures

The following grant information was disclosed by the authors:
The Deanship of Scientific Research at King Khalid University.
Kingdom of Saudi Arabia funded this work through research groups program under the project number RGP-1/103/40.

### Competing Interests

The authors declare there are no competing interests.

### Author Contributions

- Hafiza Mamona Nazir performed the experiments, analyzed the data, contributed reagents/materials/analysis tools, prepared figures and/or tables.
- Ijaz Hussain conceived and designed the experiments, authored or reviewed drafts of the paper, approved the final draft.
- Ishfaq Ahmad authored or reviewed drafts of the paper, proof reading.
- Muhammad Faisal and Ibrahim M. Almanjahie authored or reviewed drafts of the paper, approved the final draft.

### Data Availability

The raw data is available as a Supplemental File.

### Supplemental Information

Supplemental information for this article can be found online at http://dx.doi.org/10.7717/peerj.7183#supplemental-information.

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
