# Peer review of "An improved framework to predict river flow time series data"

_PeerJ, doi:10.7717/peerj.7183_

## Round 0.1 · original submission · Major Revisions

Thanks for your patience with the unusually long review process - two reviews have now been submitted.

Both reviews are very detailed and are indicating that major revisions to the manuscript are required. These revisions can be summarised as a requirement for much better, clearer and detailed justification of the methods selected, including a discussion of possible alternatives. Since you are proposing a new framework, I believe that this is an essential element of such a proposal, and will therefore greatly enhance your paper.

Secondly, Reviewer 1 points out that you are only testing your method on river flow data - but are presenting it as a generalised approach. You have two options here: either test the method on a range of other hydrological timeseries data, or focus the title - i.e. change "hydrological time-series data" to "river flow time-series data". I would recommend the second approach. I also agree with the reviewer that a much longer time-series is required for a more rigorous assessment. I would add that both the study sites appear to have some seasonal regularity, and recommend that additional sites are used - particularly those which are more complex.

In summary, these comments point to a need to improve the focus of the paper and and increase the rigor of the methodological justification and verification. I look forward to seeing your revised manuscript.

·

Basic reporting

There are several English and grammatical mistakes which should be addressed in the revised manuscript. Some of them are as follows.

--Line 34-35: Authors are suggested to differentiate hydrological time series data and meteorological factors. For example, the authors mentioned rainfall as meteorological factors. Is rainfall not a hydrological data? What is meant by horological data? Authors need to provide references otherwise. What is underlying surface? It might be changes in underlying surface or underlying surface changes.

Line 37-38: Revised by making connections from literature. For example, authors are suggesting reformulation as “In past research, several hybrid models have been developed to address these issues. In addition, explain more about the hybrid model (mention their names and application).

--Line 37-38: revise as a hybrid model “uses” instead “used”. In addition, citations need to be refined and relevant to study/result. Authors are suggested to replace with:

Zhang, X., Peng, Y., Zhang, C., & Wang, B. (2015). Are hybrid models integrated with data preprocessing techniques suitable for monthly streamflow forecasting? Some experiment evidences. Journal of Hydrology, 530, 137-152.

-needs connection between lines 38-39 and 40-42.

--Line 40-48: need more compact and concise literature instead.

--Line 42: What is the difference between multi-scale component and time scale components? In addition, the authors did not mention where and whom derived the time scale components? Please reformulate sentence.

--Line 42-44. Too long sentence: need reformulation.

--Line 46-49 # Citation is required.

--Line 49: Incorrect citation format: Campisi-Pinto et al. [8] forecasted

--Line 49-53 Revised the sentence. In addition reference style is wrong. Please check.

--Line 55-59: The findings are already published. I suggest removing or make it concise.

--Line 49-60 Revised the sentence. In addition reference style is wrong. Please check.

--Line 49-53: Revise the statement. Reference style is wrong. Too poor English is observed. The whole paper requires proofreading from native English speakers before publication.

--Line 79-83. Too long sentence. Please reformulate it.

–Line 108. Please reformulate it.

-- There are a lot of grammatical mistakes, for example, in line 167-169, what is key, I think some thing is missing.... like key importance??? What is smother? See also 172, 263-266, 306-310, 310-313, and 335-339 also. Authors should need to proofread carefully.



--Line 212-2014: Please check English and Grammer.

--Line 239-243: Similarity issue. It should be rephrased and cited accordingly.

-- Abbreviations need to be defined and used properly. In many places abbreviations are not defined, replicated and wrongly defined. For example, in line 2010, what is ANN?

Line 282. Figure 1. Wrong citation of figures. Need proper and accurate indexing and citation of figures/ tables thorought the Paper.

Experimental design

--Line: 85-86: Need consistency of between the title of manuscript and the aim of research. Are you developing an estimator? Or suggesting a predictive framework? The,title should be revised accordingly.

Line 180-181: By inspecting the nature of IMFs, authors select Laplace distribution. Where are the results of the inspection? How author inspect the nature of IMF and determine the distribution before application and case study? It needs a strong argument to support the rationale for the proposed method.

--Line 192-194: Authors should need to mention these estimators in the earlier part of the manuscript.

Validity of the findings

-- In line line 176: The authors assumed that IMFs follows a normal distribution before proposing. In the case study, I did not see the statistical result proving normality behavior of IMFs.

-- There is much difference among SNR values. For example, Table 1, CEEMDAN-EBT (the proposed method) gives very high values of SNR. Parallel to this, there is too much difference in the existing method, for example, in Kabul Inflow, EEMD-HT gives -2.4276, while EEMD-ST gives -32.0408.

That is why my major concerns are related to the validity of the assumptions, the length and type of time series being used. In addition,to assess the efficiency of the proposed methods, too much attentions are required on literature review and appropriate selection about existing methods.

Additional comments

To resolve multi-scale and noisy complexity in time series data, the authors adopted an empirical Bayesian threshold estimator in CEEMDAN and compared it with existing methods. My first concern is that whether author giving a new framework or proposing a new method? It should be clear and consistent in title, objective, methodology and in other parts of the text.
Secondly, the authors stress on hydrological time series, but used only river data. What is multi-scale? Give a clear background of multi-scale and noisy complexity in hydrological time series particularly. How author assessed the multi-scale and noisy complexity in river inflow data which they are used? This really needs attention. In addition, instead of using only river data, the authors are suggested to include results from other hydrological related time series before generalizing their method/approach. Otherwise, reduce the scope of the method by substantial text changing and literature particularly for river deta.
Thirdly, the authors are suggested to include statistical result (P-value) proving non-stationary, non-linearity before working or proposing a method / framework for handling multi-scale and noisy complexity (see lines 276). In section Proposed method (line 180), how authors inspect IMF and proposed Laplace distribution before working with data?
Fourthly, I would recommend adopting a more focused title and to increase the length of time series data for the better evaluation.

Reviewer 2 ·

Basic reporting

1. The manuscript presents a novel framework to improve prediction from inherently complex hydrological time series data, which is interesting. The subject addressed is within the scope of the journal.

Experimental design

It appears ok.

Validity of the findings

It appears ok.

Additional comments

2. However, the manuscript, in its present form, contains several weaknesses. Appropriate revisions to the following points should be undertaken in order to justify recommendation for publication.
3. For readers to quickly catch your contribution, it would be better to highlight major difficulties and challenges, and your original achievements to overcome them, in a clearer way in abstract and introduction.
4. It is mentioned in p.1 that a framework comprising complete ensemble empirical mode decomposition with adaptive noise-empirical Bayesian threshold is adopted to improve prediction from inherently complex hydrological time series data. What are other feasible alternatives? What are the advantages of adopting this particular framework over others in this case? How will this affect the results? The authors should provide more details on this.
5. It is mentioned in p.3 that an empirical Bayesian threshold estimator is adopted to optimally reduce noises from time-scale components. What are other feasible alternatives? What are the advantages of adopting this particular estimator over others in this case? How will this affect the results? The authors should provide more details on this.
6. It is mentioned in p.7 that Laplace distribution is adopted Intrinsic Mode Functions. What are other feasible alternatives? What are the advantages of adopting this particular distribution over others in this case? How will this affect the results? The authors should provide more details on this.
7. It is mentioned in p.7 that marginal maximum likelihood approach is adopted to estimate weights. What are other feasible alternatives? What are the advantages of adopting this particular approach over others in this case? How will this affect the results? The authors should provide more details on this.
8. It is mentioned in p.8 that soft threshold, hard threshold and improved threshold function are adopted as benchmark for comparison. What are the other feasible alternatives? What are the advantages of adopting these particular thresholds over others in this case? How will this affect the results? More details should be furnished.
9. It is mentioned in p.9 that “…To train model, 80% data is used, remaining 20% data is used to test the accuracy of models.…” That means no cross-validation is made. How can the problem of overfitting be resolved then? More justification should be furnished on this.
10. It is mentioned in p.9 that multi-layer perceptron neural network is adopted for the prediction of denoised IMFs. What are other feasible alternatives? What are the advantages of adopting this particular soft computing technique over others in this case? How will this affect the results? The authors should provide more details on this.
11. It is mentioned in p.9 that logistic function is adopted as activation function. What are other feasible alternatives? What are the advantages of adopting this particular activation function over others in this case? How will this affect the results? The authors should provide more details on this.
12. It is mentioned in p.9 that back-propagation algorithm which has the drawbacks of local convergence and slowness is adopted. Some justifications should be furnished on why other algorithms are not selected.
13. It is mentioned in p.10 that an autoregressive moving average model is adopted to predict the noise free IMF’s and residual. What are other feasible alternatives? What are the advantages of adopting this particular model over others in this case? How will this affect the results? The authors should provide more details on this.
14. It is mentioned in p.10 that four daily rivers inflow time series data of the Indus Basin System are adopted as case studies. What are other feasible alternatives? What are the advantages of adopting these particular case studies over others in this case? How will this affect the results? The authors should provide more details on this.
15. It is mentioned in p.11 that four statistical criteria are adopted to gauge the performance of the proposed model. What are the other feasible alternatives? What are the advantages of adopting these particular evaluation metrics over others in this case? How will this affect the results? More details should be furnished.
16. Some key parameters are not mentioned. The rationale on the choice of the particular set of parameters should be explained with more details. Have the authors experimented with other sets of values? What are the sensitivities of these parameters on the results?
17. Some assumptions are stated in various sections. Justifications should be provided on these assumptions. Evaluation on how they will affect the results should be made.
18. The discussion section in the present form is relatively weak and should be strengthened with more details and justifications.
19. There are some occasional grammatical problems within the text. It may need the attention of someone fluent in English language to enhance the readability.
20. Moreover, the manuscript could be substantially improved by relying and citing more on recent literatures about contemporary real-life case studies of soft computing techniques in hydrologic engineering such as the followings:
 Yaseen, Z.M., et al., “An enhanced extreme learning machine model for river flow forecasting: state-of-the-art, practical applications in water resource engineering area and future research direction,” Journal of Hydrology 569: 387-408 2019.
 Taormina, R., et al., “Neural network river forecasting through baseflow separation and binary-coded swarm optimization”, Journal of Hydrology 529 (3): 1788-1797 2015.
 Moazenzadeh, R., et al., “Coupling a firefly algorithm with support vector regression to predict evaporation in northern Iran,” Engineering Applications of Computational Fluid Mechanics 12 (1): 584-597 2018.
 Wu, C.L., et al., “Rainfall-Runoff Modeling Using Artificial Neural Network Coupled with Singular Spectrum Analysis”, Journal of Hydrology 399 (3-4): 394-409 2011.
 Ghorbani, M.A., et al., “Forecasting pan evaporation with an integrated Artificial Neural Network Quantum-behaved Particle Swarm Optimization model: a case study in Talesh, Northern Iran,” Engineering Applications of Computational Fluid Mechanics 12 (1): 724-737 2018.
 Chau, K.W., et al., “Use of Meta-Heuristic Techniques in Rainfall-Runoff Modelling” Water 9(3): article no. 186, 6p 2017.
21. Some inconsistencies and minor errors that needed attention are:
 Replace “…proposed a novel framework comprised on Complete Ensemble Empirical Mode Decomposition…” with “…proposed a novel framework comprising Complete Ensemble Empirical Mode Decomposition …” in lines 11-12 of p.1
 Replace “…to cope up the simple…” with “…to cope with the simple…” in line 76 of p.3
 Replace “…method is comprised on four stages…” with “…method is comprised of four stages…” in line 93 of p.4
 Replace “…method is proposed by [12] is used…” with “…method proposed by [12] is used…” in line 108 of p.4
 Replace “…it must belongs to…” with “…it must belong to…” in line 179 of p.7
 and many more…
22. In the conclusion section, the limitations of this study, suggested improvements of this work and future directions should be highlighted.

---

## Round 0.2 · Minor Revisions

I am happy to see that your manuscript has improved greatly, as also noted by both reviewers. I will be happy to see it go forward for publication, subject to just a few minor, mostly typographic changes which are needed - these should be quick to go through:

1. Title and elsewhere: "the rivers inflow" -> change to "river flow". "Inflow" implies a boundary condition; "river" (singular) is used as a category, no need for "the". This style should be adopted throughout. e.g. Abstract, L16: "rivers inflow" -> "river flow"
2. L157: "EMD technique have" -> "EMD techniques have" (pluralise)
3. L285: "cusecs" -> specify in unit notation, i.e. ft/s or ft s^{-1}. I note the use of this non-SI unit specification is acceptable in context since it is the unit of the data collection authority.
4. As suggested by Reviewer 1, please enhance the quality of labels in the graphs that are presented in figure 5, 6, 7 and 8. Units are required in these labels. Also specify the unit of time for each.
5. I suggest that you add a note at the end of the discussion with regard the seasonality of the large rivers selected, leading to some regularity in flow, and add a comment that additional testing on smaller, temperate catchments with high flow variability should be completed.

Congratulations on a high quality manuscipt.

·

Basic reporting

Authors have addressed all of my comments and resolve issues in the manuscript structure such as appropriateness of title with respect to data used in this paper, grammar mistakes, reference style, citation, text flow, etc. I am very happy to see the improved version of this paper. I am thankful to the authors to consider my comments relevant to the data and their types. From my side, the paper is fully ready to publish under the PeerJ forum. However, a little improvement if possible can be made to enhance the quality of labels in the graphs that are presented in figure 5, 6, 7 and 8. It will surely strengthen the reading quality of the paper.

Experimental design

I agree with the responses of authors which are given against my previous review.

Validity of the findings

The authors have now provided sufficient answer to all those questions which I raised in the first review of this paper.

Additional comments

Congratulations! The manuscript has been improved considerably. I recommend this paper for its publication in PeerJ forum. However, I suggested improving the label of figure 5, 6, 7, and 8.

Reviewer 2 ·

Basic reporting

ok

Experimental design

ok

Validity of the findings

ok

Additional comments

The revised paper has addressed all my previous comments, and I suggest to ACCEPT the paper as it is now.

---

## Round 0.3 · accepted · Accept

Thanks for making the changes and including the additional paragraph. I look forward to seeing the final version in PeerJ.